# Clinical Outcomes and Joint Stability after Lateralized Reverse Total Shoulder Arthroplasty with and without Subscapularis Repair: A Meta-Analysis

**DOI:** 10.3390/jcm10143014

**Published:** 2021-07-06

**Authors:** Katia Corona, Simone Cerciello, Gianluca Ciolli, Lorenzo Proietti, Riccardo D’Ambrosi, Adriano Braile, Giuseppe Toro, Alfonso Maria Romano, Francesco Ascione

**Affiliations:** 1Department of Medicine and Health Sciences “Vincenzo Tiberio”, University of Molise, Via Giovanni Paolo II, 86100 Campobasso, Italy; 2Department of Orthopaedics, Agostino Gemelli University Hospital Foundation IRCCS, Catholic University, 00168 Rome, Italy; simone.cerciello@policlinicogemelli.it (S.C.); gianluca.ciolli01@icatt.it (G.C.); 3Casa di Cura Villa Betania, 00165 Rome, Italy; proiettilorenzo@hotmail.com; 4Marrelli Hospital, 88900 Crotone, Italy; 5IRCCS Istituto Ortopedico Galeazzi, 20161 Milano, Italy; riccardo.dambrosi@hotmail.it; 6Multidisciplinary Department of Medico-Surgical and Dentistry Specialties, University “Campana Luigi Vanvitelli”, 81100 Napoli, Italy; adrybrail@hotmail.it (A.B.); giuseppe.toro@unicampania.it (G.T.); 7Department of Orthopaedic and Trauma Surgery, Ospedale Buon Consiglio Fatebenefratelli, 80123 Napoli, Italy; alfonso.maria.romano@gmail.com (A.M.R.); francescoascione.md@gmail.com (F.A.); 8Orthopedics and Sport Medicine Unit, Campolongo Hospital, 84127 Salerno, Italy

**Keywords:** lateralized reverse shoulder arthroplasty, subscapularis repair, clinical outcomes, dislocation, complication rates, meta-analysis

## Abstract

Introduction: Subscapularis tendon repair in reverse total shoulder arthroplasty represents a potentially modifiable risk factor for dislocation, and its role continues to be debated. The purpose of the present meta-analysis was to compare the outcomes of the primary lateralized RSAs with and without subscapularis repair in terms of range of motion, clinical outcomes, dislocations, and complications rate. Materials and Methods: A systematic literature search in MEDLINE (Pubmed), Embase, and the Cochrane Central Register of Controlled Trials database was carried up to December 2020. A data extraction form was developed to collect select data from the included studies. The methodological quality was assessed using a Methodological Index for Nonrandomized Studies (MINORS) score. Statistical analysis was performed with Review Manager (Version 5.4, The Cochrane Collaboration). Results: A total of four comparative studies involving 978 patients were included. In the pooled analysis, the reinsertion of the subscapularis yielded better functional outcomes in terms of the constant (P < 0.00001) and ASES (P = 0.002) scores. The forward elevation, external rotation at 0°, internal rotation, and dislocation rates were comparable between the two groups (P = n.s.), while statistically increased abduction was observed in those patients who did not have their subscapularis repaired (P < 0.00001). Conclusion: The results of the present findings suggest that it seems reasonable to reinsert the subscapularis whenever it is present, in good tissue conditions, and with no evidence of fatty degeneration of its muscle belly. Level of evidence: Level III meta-analysis

## 1. Introduction

Subscapularis (SSc) tendon repair has become a contentious matter in reverse total shoulder arthroplasty (RSA), with increasing debate surrounding the conflicting results concerning biomechanics [1,2,3,4,5] and clinical outcomes [6,7,8,9,10,11]. The choice to repair the SSc tendon or not usually depends on the surgeon’s inclination and judgment. The surgeons who favor its repair highlight its role in increasing joint stability and strength in internal rotation and in avoiding large empty spaces, which may potentially lead to increased rates of infection [12]. Conversely, other surgeons argue that reinserting the SSc limits external rotation and abduction and is biomechanically adverse for deltoid function, as the SSc functions as an adductor in the RSA, necessitating the deltoid function to facilitate the raising of the arm [13,14,15].

Additionally, an SSc tendon repair needs to be protected, and delayed postoperative rehabilitation requires it. The external rotation has a major impact on the functioning of the arm as one goes about their daily chores and activities. Therefore, delaying this function, even if temporarily, may adversely affect the patient’s quality of life, especially for the elderly patients.

In any case, the effect of the SSc reinsertion is variable and depends on the choice of implants, since they have different designs. The moment arms of the rotator cuff and deltoid are significantly altered when the center of rotation (COR) is medialized or lateralized [12,16,17]. Biomechanically, a lateralized design can increase joint loads, owing to the increase of the impingement-free range of motion (ROM) and the deltoid force required during abduction [2,3,4,5,17].

A recent meta-analysis [18] on the role of the SSc reinsertion on the postoperative dislocation rate confirmed its protective function but did not show any difference between lateralized or medialized designs. Furthermore, when the SSc tendon was not repaired, a design with lateralized glenosphere proved to be protective against dislocation [18]. The question of whether to repair the SSc or not in the RSAs continues to challenge surgeons. More recently, other studies have been published in the literature pertaining to the matter, especially with regard to the utilization of implants with a lateralized design.

Thus, the aim of the present meta-analysis was to update the available literature by evaluating the efficacy of the SSc repair in patients that underwent primary lateralized RSAs, on the clinical outcomes, ROM, dislocations, and complications rate. We hypothesized that the reinsertion of the SSc when a lateralized design was used is biomechanically unfavorable in terms of its clinical and functional outcomes.

## 2. Materials and Methods

This systematic review was conducted according to the Preferred Reporting Items for Systematic Reviews and Meta-analyses (PRISMA) guidelines [19].

### 2.1. Search Strategy and Study Selection Process

A systematic electronic search of the following databases was performed in December 2020: Pubmed-Medline, Embase, and the Cochrane Central Register of Controlled Trials database. The keywords were “subscapularis tendon”, “repair”, “arthroplasty”, “reverse shoulder arthroplasty”, “inverse arthroplasty”, and “lateralized”, which were used with the Boolean operators, “AND” or “OR”.

Two specialists in shoulder surgery (A.B. and G.T.) independently reviewed the title and the abstract of each article from the literature search. The full text of an article was obtained and evaluated when its eligibility could not be gauged during the first screening. Any disagreements that arose were resolved through a discussion between the reviewers, and a third reviewer (F.A.) was consulted if the disagreement could not be resolved. The reference lists of the included studies and the relevant systematic reviews were manually searched in case any potentially significant studies were overlooked.

### 2.2. Eligibility Criteria

The following types of studies were considered for inclusion: (1) those that included patients undergoing a primary lateralized RSA; (2) those that included patients for whom subscapularis repair had been performed (trial group) as compared to patients for whom the subscapularis was not repaired (control group); (3) a published randomized control trial (RCT) or no RCT, retrospective (RE), or prospective (PRO) trials. All non-English and review articles, animal experiments, and in vitro trials were excluded from the review analysis.

### 2.3. Data Extraction

An electronic piloted form was generated for data extraction. For each eligible study, data on patient demographics (age, gender, follow-up, type of prosthesis used), indication for surgery, postoperative outcomes (abduction, forward elevation, external rotation at 0° and internal rotation), constant, and ASES score and dislocations and complications were extracted and recorded by two authors independently.

### 2.4. Evaluation of the Quality of Studies

The quality was assessed using the Methodological Index for Nonrandomized Studies (MINORS) [20,21]. This is a validated tool for the methodological assessment of non-randomized surgical studies, whether comparative or non-comparative. For comparative studies, 12 criteria are used, and items are scored from 0 to 2: where 0 = not reported, 1 = reported but inadequate, and 2 = reported and adequate. The global ideal score for comparative studies is 24. This evaluation was carried out by two authors (A.B. and G.T.), and it included a debate to reach a consensus in the event of disagreement.

## 3. Statistical Analysis

Review Manager (Version 5.3, The Cochrane Collaboration) was adopted to estimate the outcomes among selected studies. For continuous variables, the mean difference was utilized, while dichotomous variables were expressed with risk ratio (RR) and odds ratio (OR). They were reported with 95% confidence intervals (95% CI), and the P value of 0.05 was used as the level of statistical significance. The statistical heterogeneity was measured through the I-square (I^2^) test, with significance set at P < 0.10. I^2^ > 40% would be defined as significant heterogeneity, which means the random effects model would be applied to pool the results. Otherwise, the fixed effects model would be applied.

## 4. Results

### 4.1. Search Results

The initial literature search identified 830 studies. After removing 267 duplicates, 563 studies remained. Of these, after reviewing the abstracts, 301 were eliminated, leaving 262 articles to be screened. An additional 225 articles were excluded after the inclusion and exclusion criteria analysis. Additional studies were not found when the reference list of the included articles was manually checked. Finally, four articles met the criteria for inclusion. The flowchart for the research selection process is shown in Figure 1. The eligible studies had a mean MINORS score of 19.5 (range, 21 to 22), indicating that the available literature was of good methodological quality (Table 1).

### 4.2. Patient and Study Characteristics

The demographic characteristics of the two cohorts of patients are shown in Table 2. A total of 978 patients were identified. The SSc repair group consisted of 535 subjects and the SSc non-repair group consisted of 443 subjects. Three studies reported information on gender. In the SSc repair group (intervention), 157 were men and 298 were woman, while in the SSc non-repair group (control), 172 were men and 157 were woman.

### 4.3. Indication for Surgery

The main indications for surgery were summarized in Table 2.

The decision to whether repair the SSc or not was given intra-operatively in all studies and was based on the surgeon’s preference and subscapularis tendon’s macroscopic appearance.

### 4.4. Range of Motion (ROM) Outcomes

Two studies reported comparative results in terms of abduction, as shown in Table 3. In the intervention group, the mean abduction increased from 68.8°_pre_ to 109.6°_post_, while in the control group, it improved from and 77°_pre_ to 123.5°_post_. The analysis of the postoperative scores showed that there was a significant difference between the two cohorts in favour of the no-subscapularis repair group MD = −12.30 (95% CI, −16.68 to −7.93; P < 0.00001) (Figure 2).

In the intervention group, the mean forward elevation increased from 68.8°_pre_ to 109.6°_post_, while in the control group, it improved from 82.9°_pre_ to 128.6°_post_. The analysis of postoperative scores showed that there was no significant difference between the two groups (MD = 3.88 (95% CI, −0.37 to 8.13; P = 0.07)) (Figure 2).

In the intervention group, the mean external rotation at 0° increased from 11.4°_pre_ to 29.9°_post_, while in the control group, it improved from and 18.1°_pre_ to 31.8°_post_. The analysis of postoperative scores denoted that there was no significant difference between the two groups (MD = −1.18 (95% CI, −3.87 to 1.51; P = 0.34)) (Figure 2).

In the intervention group, the mean internal rotation increased from 3.3°_pre_ to 5.1°_post_, while in the control group, it improved from 3.2°_pre_ to 4.4°_post_. The analysis of postoperative scores showed that there was significant difference between the two groups in favor of the subscapularis repair group (MD = 0.68 (95% CI, 0.46 to 0.89; P < 0.00001)) (Figure 2).

### 4.5. Clinical Outcomes

The analysis of postoperative constant scores showed that there was significant difference between the two groups in favor of the subscapularis repair group (MD = 4.81 (95% CI, 2.63 to 6.98; P < 0.00001)) (Figure 3).

The analysis of postoperative ASES scores revealed that there was significant difference between the two groups in favor of the subscapularis repair group (MD = 4.13 (95% CI, 2.63 to 6.98; P = 0.002)) (Figure 3).

### 4.6. Dislocation and Complication Rates

All studies reported dislocation and complication rates, as shown in Table 4. The analysis of the dislocation rate indicated that there was no significant difference between the two groups (OR = 0.31 (95% CI, 0.08 to 1.15; P = 0.08)) (Figure 4).

The analysis of the complication rate implied that there was no significant difference between the two groups (OR = 1.10 (95% CI, 0.64 to 1.89; P = 0.74)) (Figure 4).

## 5. Discussion

The most important finding in the present meta-analysis is that the reinsertion of the SSc when a lateralized RSA was used does not seem to play a key role in maintaining joint stability and achieving better clinical outcomes.

With data pooling, a lateralized RSA with the reinsertion of the SSc yielded no difference in terms of forward elevation, external rotation at 0°, and internal rotation when compared to the cohort in which the SSc was not reinserted. Conversely, the active abduction was statistically superior in the cohort in which the SSc was not reinserted. Regarding the constant and ASES score, the group for which the SSc was repaired reported a statistically significant difference when compared to the cohort in which the SSc was not reinserted.

Given the paucity of the studies published in the literature and their heterogeneity, the role of SSc tendon repair in a lateralized RSA is still being debated. This is the first meta-analysis that analyzes this aspect. It takes into consideration only those studies that utilized a lateralized prosthetic design, including four retrospective studies that reported contrasting findings. Specifically, ROM was not different between the two groups in three studies, while the other two reported improved internal rotation among repaired patients with the SSc repaired and improved abduction among patients for whom the SSc was not reinserted. There were no differences in the clinical scores reported in three out of the four retrieved articles.

Generally, in biomechanical models of the RSA, the SSc tendon has been portrayed as an adductor counteracting the deltoid forces, especially during the first 70° of abduction [3,13]. This aspect potentially limits external rotation and abduction function after the SSc reinsertion. The last concept has been confirmed by the one study [8], since the active abduction was statistically superior in the cohort in which the SSc was not reinserted. On the contrary, no difference emerged between the two groups in terms of external rotation at 0° in any of the included studies.

In the existing literature, there are few studies analyzing the combined effects of lateralized RSAs and the SSc, repaired or not repaired, on the clinical outcomes. A recent systematic review [18] reported that implants with lateralized glenosphere seem to be protective against dislocation when the SSc tendon is not repaired with dislocation rates being significantly lower in this cohort. Although it is evident that a lateralized design increases residual cuff tension with increased joint stability independently of the SSc reinsertion, our findings did not match with those of Matthewson et al. [18]. In fact, similar complication and dislocation rates were observed in the two groups.

Furthermore, it is appropriate to take into account that it is possible to reinsert the SSc unless the tissue quality is degenerated or atrophic, or the muscle belly shows severe fatty infiltration, or the tendon is torn and retracted. In these scenarios, the choice of a more lateralized implant might be preferable to a medialized implant [18].

There are several relevant limitations to this study. First, the number of published studies found is very limited, with one large study having a major weight on the pooled results, resulting in a conclusion based only a single study. The main limitations in the current meta-analysis of a rare event, such as dislocation, were given by the limited number of available studies with their high heterogeneity and missing data.

Secondly, all studies were retrospective, and the patients were not randomized according to the subscapularis management. The surgeon decided to repair the subscapularis based on his preference, the quality of the tendon, the ease with which the tendon could be brought back to its original footprint, and the quality of the smaller tubercle. This limitation may have resulted in a bias in the selection of patients.

Another weakness of the present meta-analysis is the lack of information on the effective healing of the tendon at the latest follow-up. The SSc reinsertion after the RSA has similar risks of failure as the other tendons after rotator cuff repair [22,23]. Therefore, not all patients having had their SSc reinserted may have at last FU.

Nonetheless, the present meta-analysis is the first in the literature to critically evaluate and quantify the effects of subscapularis repair on function, ROM, and stability after lateralized RSAs.

The clinical relevance of the present meta-analysis was that the findings provide orthopedic surgeons with important information on the decision-making criteria for the management of subscapularis tendon in the setting of a lateralized RSA. Given the relevant potential biases in our meta-analysis, more adequately powered and better-designed RCT studies with long-term follow-up would be required to reach a more robust conclusion.

## 6. Conclusions

According to the present findings, the reinsertion of the SSc seems to be irrelevant in maintaining joint stability and achieving better clinical outcomes when a lateralized prosthetic design was utilized. It seems reasonable to reinsert the SSc whenever it is present, in good tissue conditions and with no evidence of fatty degeneration of its muscle belly. Otherwise, it is recommended not to repair the SSc considering that a lateralized implant theoretically improves stability.

## Figures and Tables

**Figure 1 jcm-10-03014-f001:**
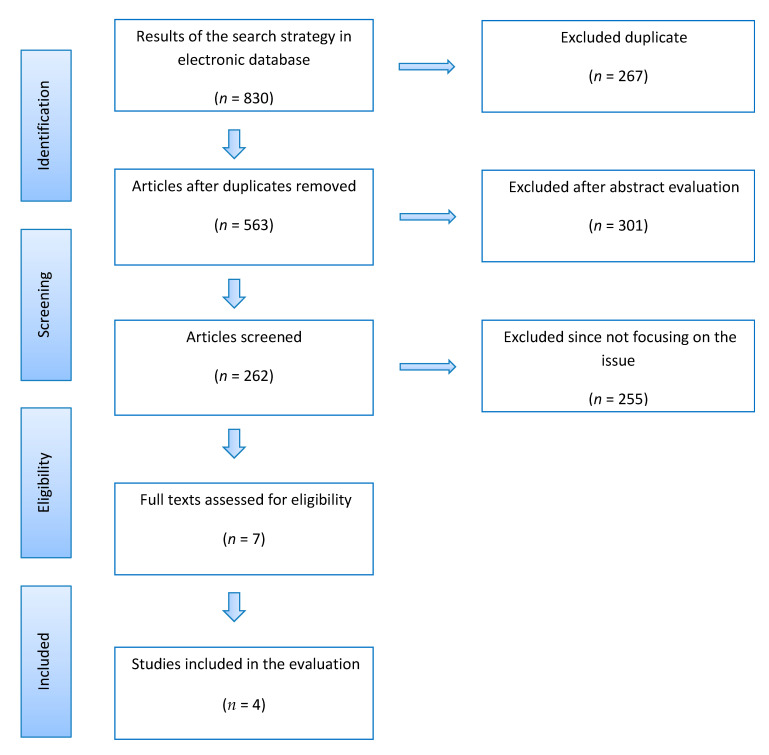
PRISMA flow chart of the paper selection process.

**Figure 2 jcm-10-03014-f002:**
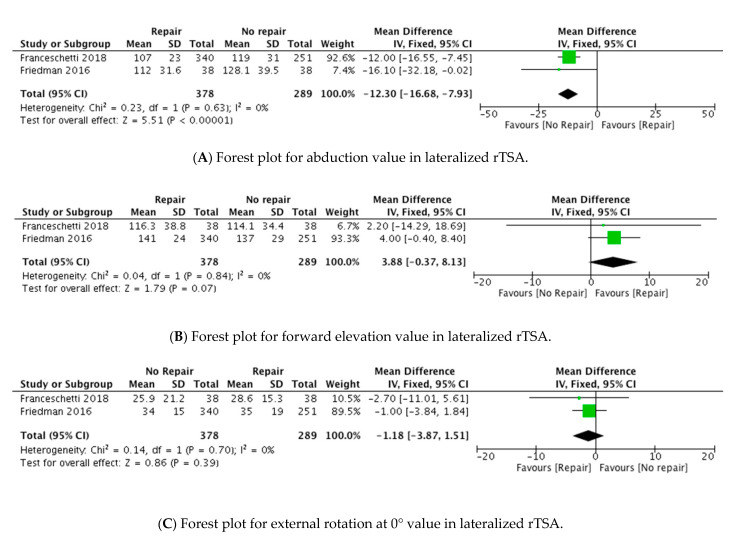
Forest plots of primary outcomes.

**Figure 3 jcm-10-03014-f003:**
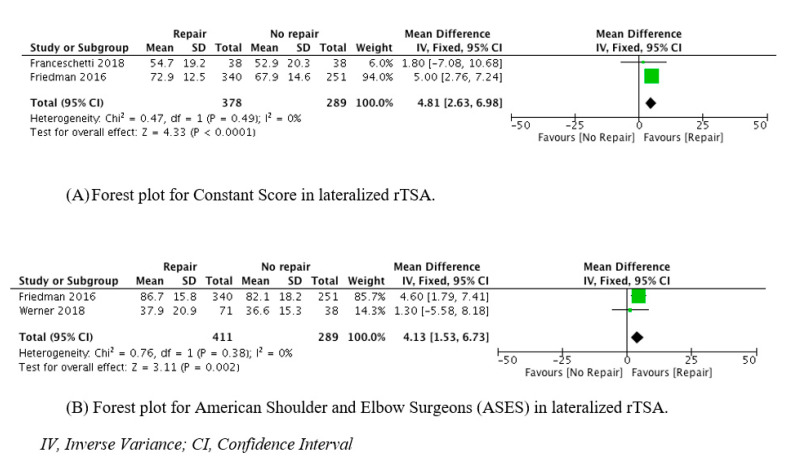
Forest plots of secondary outcomes.

**Figure 4 jcm-10-03014-f004:**
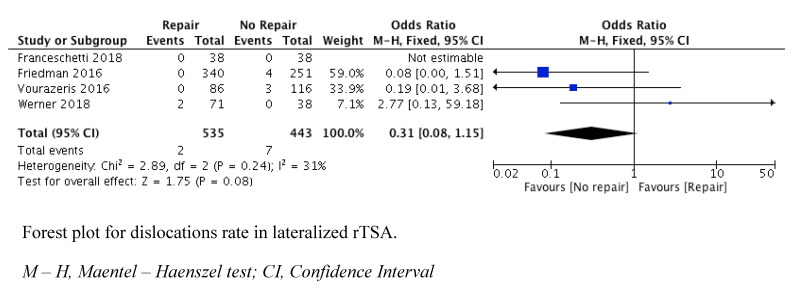
Forest plots of dislocations rates.

**Table 1 jcm-10-03014-t001:** Methodological assessment of retrieved articles (MINORS) score of the included studies.

	Vourazeris, 2017	Friedman, 2017	Werner, 2018	Franceschetti, 2019
Clearly stated aims	2	2	2	2
Consecutive patients	2	1	2	2
Prospective data collection	0	0	0	2
Appropriate endpoints	2	2	2	2
Unbiased assessment	2	2	2	2
Appropriate follow-up	2	2	2	2
<5% lost to follow-up	0	2	2	2
Prospective study size calculation	0	0	0	0
Adequate control group	2	2	2	2
Contemporary groups	2	2	2	2
Baseline comparability	2	1	2	2
Adequate statistic	2	2	2	2
Total	18	18	20	22

**Table 2 jcm-10-03014-t002:** Demographics of the cohort reviewed.

Author	Journal	Group	Number	M/F	Age Mean (SD)	FU Mean (SD)	Prosthetic Design	Indications for Surgery
Vourazeris, 2016	JSES	SR NR	86 116	-	71.6 71.1	39.6 37.2	MGLH+ *	Cuff tear arthropathy RCT with OA Rheumatoid arthritis Avascular necrosis of the humeral head with RCT
Friedman, 2016	JSES	SR NR	340 251	119/221 146/105	72.9 71.9	37.3 35.7	MGLH+ *	Cuff tear arthropathy RCT with OA Rheumatoid arthritis
Werner, 2018	JAAOS	SR NR	71 38	28/43 15/23	71.1 (10.7) 70.7 (8.6)	24 (18) 25.2 (18)	LGLH+ **	Cuff tear arthropathy RCT with OA
Franceschetti, 2018	Int Ort	SR NR	38 38	10/34 11/29	70.1 (10.6) 69.7 (6.1)	15.9 (1.2) 16.9 (1.9)	MGLH+ **	Cuff tear arthropathy

SD, Standard Deviation; FU, Follow-Up. Int Ort, International Orthopaedics. JSES, Journal of Shoulder and Elbow Surgery; JAAOS, Journal of American Academy of Orthopedic Surgeons; SR; Subscapularis Repair; NR; No Subscapularis Repair; M, Men; F, Females; MGLH, Medialized Glenoid Lateralized Humerus; LGLH, Lateralized Glenoid Lateralized Humerus; OA, Osteoarthritis; RCT, Rotator Cuff Tear. * Equinoxe (Exatech). ** Aequalis Ascend ^TM^ Flex (Tornier—Wright).

**Table 3 jcm-10-03014-t003:** Post-operative range of motion and clinical outcomes.

	Vourazeris, 2016		Friedman, 2016		Franceschetti, 2018		Werner, 2018	
	SR	NR	SR	NR	SR	NR	SR	NR
Range of motion								
Abduction (°) mean (SD)	109	122	107 (23)	119 (31)	112.2 (31.6)	128.1 (39.5)		
Forward flexion (°) mean (SD)			141 (24)	137(29)	116.3 (38.8)	114.1 (34.4)		
External rotation at 0° (°) mean (SD)	24	26	34 (15)	35 (19)	25.9 (21.2)	28.6 (15.3)		
Internal rotation (°) mean (SD)			5.1 (1.3)	4.4 (1.6)	5.1 (1.28)	4.5 (0.9)		
Clinical outcomes								
Constant mean (SD)	72.6	72.7	72.9 (12.5)	67.9 (14.6)	54.7 (19.2)	52.9 (20.3)		
ASES mean (SD)	77.7	79.3	86.7 (15.8)	82.1 (18.2)			37.9 (20.9)	36.6 (15.3)

SR, Subscapularis Repair; NR, No Subscapularis Repair; ASES, America Shoulder and Elbow Score.

**Table 4 jcm-10-03014-t004:** Comparison of dislocation and complications rate for subscapularis repair and subscapularis non-repair cohorts.

	Vourazeris, 2016		Friedman, 2016		Franceschetti, 2018		Werner, 2018	
	SR	NR	SR	NR	SR	NR	SR	NR
Range of motion								
Abduction (°) mean (SD)	109	122	107 (23)	119 (31)	112.2 (31.6)	128.1 (39.5)		
Forward flexion (°) mean (SD)			141 (24)	137(29)	116.3 (38.8)	114.1 (34.4)		
External rotation at 0° (°) mean (SD)	24	26	34 (15)	35 (19)	25.9 (21.2)	28.6 (15.3)		
Internal rotation (°) mean (SD)			5.1 (1.3)	4.4 (1.6)	5.1 (1.28)	4.5 (0.9)		
Clinical outcomes								
Constant mean (SD)	72.6	72.7	72.9 (12.5)	67.9 (14.6)	54.7 (19.2)	52.9 (20.3)		
ASES mean (SD)	77.7	79.3	86.7 (15.8)	82.1 (18.2)			37.9 (20.9)	36.6 (15.3)

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
