# Peer review of "Clinical Outcomes and Joint Stability after Lateralized Reverse Total Shoulder Arthroplasty with and without Subscapularis Repair: A Meta-Analysis"

_jcm, 2021, doi:10.3390/jcm10143014_

Round 1
Reviewer 1 Report
The authors have revised their systematic review adequately and I am overall satisfied by the responses. The manuscript improved as a result. Few remaining minor issues can be edited:
- In table 1, delete "[range]" in the header since none are presented
- Lines 136, 141, 145, 149: replace "study" by "intervention". The repetitive text formatting makes reading suboptimal. Alternative wordings between sections may help.
- Table 3: it is still puzzling that the study of Vourezeris do not present SDs in their results, but this is a fact. The authors should attempt to contact the authors to get these measures of dispersion (SDs), and report it, or use their data in the pooling.
For the ROM, there can be also mention that the data are mean and SD .. not only for Constant and ASES.
Why do we have such huge difference in ASES between the 2 studies from Friedman and Werner? - Figure 2: correct "extrarotation" and "intrarotation" with the adequate wording (see previous comments)
- Figure 4: I recommend to remove the forest plot for the complication rates. Table 4 is sufficient to present these data. I do not think pooling to be justified because of indetermined AE definitions.
- Lines 175-177: this is moved into the discusison and therefore here a repetition.
- Lines 178-179: this sentence seems wrong and may be better removed
- Line 190: format reference [Friedman]
- Line 228: delete "therefore"
Author Response
Response to the review 1
The authors have revised their systematic review adequately and I am overall satisfied by the responses. The manuscript improved as a result. Few remaining minor issues can be edited:
- In table 1, delete "[range]" in the header since none are presented
- Response: we have deleted "[range]" in the header, as suggested
- Lines 136, 141, 145, 149: replace "study" by "intervention". The repetitive text formatting makes reading suboptimal. Alternative wordings between sections may help.
- Response: we have replaced "study" by "intervention" in the entire manuscript, as suggested
- Table 3: it is still puzzling that the study of Vourezeris do not present SDs in their results, but this is a fact. The authors should attempt to contact the authors to get these measures of dispersion (SDs), and report it, or use their data in the pooling.
For the ROM, there can be also mention that the data are mean and SD not only for Constant and ASES. - Response: we proceeded to send an email at “jvourazeris@orthoep.com” on June 24, 2021 but without any response. Unfortunately, we only have three days to submit the revision.
- Figure 2: correct "extrarotation" and "intrarotation" with the adequate wording (see previous comments)
- Response: We have corrected extrarotation" and "intrarotation", as suggested
- Figure 4: I recommend to remove the forest plot for the complication rates. Table 4 is sufficient to present these data. I do not think pooling to be justified because of indetermined AE definitions.
- Response: We have removed the forest plot for the complication rates, as suggested.
- Lines 175-177: this is moved into the discussion and therefore here a repetition.
- Response: we have removed this sentence and re-worked in the discussion.
- Lines 178-179: this sentence seems wrong and may be better removed
- Response: we have removed this sentence.
- Line 190: format reference [Friedman]
- Response: we have entered reference [8]
- Line 228: delete "therefore"
- Response: we have deleted "therefore", as suggested
Reviewer 2 Report
Authors thoroughly addressed the issues highlighted in the previous review, improving the quality of their manuscript.
Author Response
.
Reviewer 3 Report
Recommend English Style Revision.
Author Response

(The authors gave the same response as above.)
